# Neurotoxicity and Other Clinical Manifestations of a Common European Adder (*Vipera berus)* Bite in Romania

**DOI:** 10.3390/toxins14070500

**Published:** 2022-07-18

**Authors:** Gabriela Viorela Nițescu, Coriolan Emil Ulmeanu, Maria-Dorina Crăciun, Alina Maria Ciucă, Alexandru Ulici, Ioan Ghira, Davide Lonati

**Affiliations:** 1Department of Pediatrics, University of Medicine and Pharmacy “Carol Davila”, 020021 Bucharest, Romania; coriolan.ulmeanu@umfcd.ro; 2Emergency Clinical Hospital for Children “Grigore Alexandrescu”, 011743 Bucharest, Romania; maria.craciun@umfcd.ro (M.-D.C.); alina-maria.ciuca@rez.umfcd.ro (A.M.C.); alexandru.ulici@umfcd.ro (A.U.); 3Department of Epidemiology, University of Medicine and Pharmacy “Carol Davila”, 020021 Bucharest, Romania; 4Department of Pediatric Surgery and Orthopedics, University of Medicine and Pharmacy “Carol Davila”, 020021 Bucharest, Romania; 5Romanian Association of Herpetology, 400006 Cluj-Napoca, Romania; ioan.ghira@ubbcluj.ro; 6Poison Control Centre and National Toxicology Information Centre, Toxicology Unit, IRCCS, Maugeri Foundation, 27100 Pavia, Italy; davide.lonati@icsmaugeri.it

**Keywords:** neurotoxicity, Berus Viper, common European viper, child, monospecific antivenom, south Romania

## Abstract

Most cases of envenomation by common European vipers (*Vipera berus*) have not been reported to have neurotoxic manifestations. However, these manifestations have been demonstrated in some cases of envenomation by subspecies of *V. berus*, found in the Carpathian Basin region of south-eastern Europe. Here, we report the case of a 5-year-old girl from the south of Romania who presented symptoms of neurotoxicity, as well as other systemic and local symptoms, after being bitten by an adder of the *V. berus* subspecies. Treatment consisted of monovalent antivenom, a corticosteroid, and prophylactic enoxaparin. Neurotoxic manifestations of envenomation as well as other local and systemic symptoms improved within 5 days of treatment. The presented case shows that venom from *V. berus* subspecies found in the Carpathian Basin can have neurotoxic effects. This case also confirmed the efficacy of monospecific antivenom treatment in bringing about rapid and complete remission, following envenomation.

## 1. Introduction

According to the Romanian Society of Herpetology, the only venomous snakes in Romania are adders (also known as vipers). These can be classified under three species, namely, *Vipera berus*, *Vipera ammodytes*, and *Vipera ursinii* [1]. Clinical manifestations of envenomation by these adders vary and may be local or systemic. Systemic manifestations of envenomation may present as gastrointestinal, cardiac, neurological, and/or respiratory symptoms, and may also include anaphylactic reactions and coagulopathy [2,3,4,5,6]. Neurotoxic manifestations are considered unusual; in most cases published in Europe, these manifestations were attributed to *V. aspis* [4,6], and not to *V. berus*. Nevertheless, clinical symptoms of neurotoxicity in *V. berus* bites were observed as far back as 2008, when Malina et al. reported a previously healthy 27-year-old man bitten by a *V. berus* in eastern Hungary [7] Later, in 2017, the same author also showed the possibility of neurotoxic manifestations of *V. berus* envenomation during an experimental study of *V. berus* venom of adders from the same region. [8]. In this case report, we document the presence of symptoms of neurotoxicity and the development of these symptoms, as well as other associated clinical manifestations of envenomation, in a 5-year-old girl from south Romania, who was bitten by a common European viper (*V. berus*). The study was conducted according to the provisions of the “EU Guidelines for the Promotion and Protection of the Rights of the Child” and was approved by the Ethics Committee of the hospital (Protocol Code 15351/02.06.2022).

## 2. Case Report

A previously healthy 5-year-old girl was bitten on her right leg by a viper near her house, in south Romania. Twelve hours later, she was presented to our clinic, after being transferred from a local hospital.

On admission, the girl presented with a mildly altered general status. Examination revealed local signs of a snake bite on her right leg, including local pain and heat, swelling, infiltration, erythema surrounded by cyanosis and functional impairment. The snake bite mark is shown in Figure 1a. Systemic manifestations were also present, and include mild gastrointestinal and neurological symptoms. Gastrointestinal symptoms consisted of diffuse abdominal pain and nausea. Neurological manifestations included severe somnolence and axial hypotonia. Moreover, bilateral ophthalmologic disturbances (shown in Figure 1b) were noted, including palpebral ptosis and ophthalmoplegia. Diplopia was also confirmed.

The initial cardiovascular assessment did not show any abnormalities; blood pressure and heart rate were within the normal ranges according to age (102/55 mmHg and 110 bpm, respectively). The results of laboratory tests were normal (see Table 1). Electrocardiogram and cardiac echography showed no abnormalities, and Doppler echography of the right lower limb showed normal results.

One hour after admission, local signs and general symptoms worsened and swelling and pain spread to the middle of the right calf, somnolence and palpebral ptosis became pronounced and blood pressure started to decrease gradually (to 80/67 mmHg, 1 h post-admission). Based on the above clinical presentation, the severity of the case was graded as 2b (regional oedema and moderate general symptoms as mild hypotension and neurotoxic signs), according to the Audebert–Boels classification, adapted to children by Marano et al. (see Table 2) [6].

The viper was identified as belonging to the species *V. berus* based on the morphological description of the viper provided by the girl’s mother and the information obtained from the Romanian Association of Herpetology. The mother who accompanied the child at the moment of the incident described a completely black snake without a zigzag pattern and about 50 cm long. The last mapping of the distribution of vipers in Romania shows the presence of *V. berus* and *V. ammodytes* in the geographical location of the incident reported as the Subcarpathian area of south Romania, specifically Vâlcea county [1]. According to the mapping, the herpetologist stated that the viper belongs to the *V. berus* species, because *V. ammodytes* does not have any melanic gene and thus cannot be entirely black. Even though some authors have reported the presence of a certain subspecies of vipers, namely, *Vipera berus nikolskii* in Vâlcea [9], it is rather difficult to establish the viper in question belonging to this subspecies whose presence here has not been certified by DNA identification.

Twenty hours after the viper bite, treatment was initiated with Viper Venom Antitoxin (*Immunoserum contra venena viperarum europaearum*) manufactured by Biomed, Poland. The patient was administered 500 AU (one vial) as a bolus dose in 250 mL normal saline over 3 h. The patient was also treated with intravenous fluid therapy containing glucose and electrolytes, intravenous methylprednisolone, and subcutaneous enoxaparin (prophylactic dosage of 2000 UI/day) for 5 days.

The patient’s progress was clinically monitored after the initiation of the treatment. The laboratory results and symptoms over time are presented below (see Table 1 and Table 3). Thirty minutes after administration of the antivenom, ocular movements reappeared and blood pressure increased to 98/66 mmHg. Somnolence, diplopia, and gastrointestinal manifestations withdrew on Day 2, while palpebral ptosis disappeared on Day 4. Local signs, after initially worsening progressively, improved and completely disappeared by Day 5.

The child was discharged on Day 6 without any clinical signs or symptoms. The laboratory test results were within normal ranges (see Table 1). No immediate or delayed adverse reactions were reported after antivenom administration, apart from a slight increase in IgE level, without clinical expression.

Daily lab testing was conducted using bedside assessment.

## 3. Discussion

From the venom composition, it is thought that neurotoxic effects of venom from common European adders are caused by neurotoxins with phospholipase A2 (PLA2) enzymatic activity [10,11]. A study published by Zanetti et al. in 2018 regarding the effects of *V. aspis* and *V. berus* venoms in mice showed that both types of venom have phospholipase A2 enzymes, but only *V. aspis* venom is neurotoxic. Clinical observations from various regions of France and Italy support this finding, attributing neurological manifestations exclusively to envenomation by *V. aspis* [4,6,12,13,14]. Nevertheless, studies and clinical observations [8,10] have shown that the venom of some *V. berus* subspecies from Eastern Europe can have neurotoxic effects, not only in animal experimental models but also in humans. Neurotoxic manifestations following *V. berus* envenomation have been sporadically communicated in the literature, with all incidents reported in the Carpathian Basin of Hungary, Romania, and Bulgaria [8,15]. Three cases out of the seven reported cases involved children aged 14, 12, and 9 years, respectively [16,17,18].

In our case, the incident took place in the Subcarpathian area of the south of Romania, which is in the geographical region of Oltenia (see Figure 2). Symptoms of neurotoxicity appeared 30 min after envenomation and included somnolence, palpebral ptosis, ophthalmoplegia, and bilateral diplopia. These symptoms are similar to symptoms previously described in the literature in cases of *V. berus* bites in the Carpathian Basin region [8]. Antivenom indication was established after classifying the case severity as grade 2b, using the Audebert–Boels classification adapted to children by Marano et al. as shown in Table 2 [6].

Using other classification systems published in the literature, regarding severity of viper envenomation, we can classify our case as Stage 2 according to the Clinical Gradation of European Viper Envenomation system [12], and Stage 1 (ocular and mild gastrointestinal symptoms present) according to the Modified Grading Severity Score (GSS) system for peripheral neurotoxic effects, after Italian viper bites [4]. Both stages in the above classification systems are stages in which antivenom treatment is indicated.

In our case, viper Venom Antitoxin serum (Biomed, Poland) was administered. Viper Venom Antitoxin serum is a monovalent antivenom containing Fab specific equine immunoglobulins that bind the venom of *V. berus.*

Enoxaparin was administered prophylactically to prevent secondary venous thrombosis due to prolonged immobilization of the affected leg [12]. We excluded the need for antibiotic therapy since there were no local or general signs of infection. As stated in the literature, antibiotics in snakebite cases should only be administered if there is a positive history of infection, or clinical or biological signs of infection. [6,19].

Although the use of corticosteroids is controversial [6,19], the decision was made to administer intravenous methylprednisolone as a symptomatic treatment for its well-known anti-inflammatory effects. The outcome of methylprednisolone was favourable. Progressive amelioration of neurological symptoms was noted as follows: 30 min after immunotherapy administration, ocular movements reappeared; after 12 h, somnolence and diplopia withdrew; and on the fourth day, palpebral ptosis was completely absent. We believe that the complete resolution of symptoms of neurotoxicity after treatment with the monospecific antivenom could be an argument that the snake responsible for envenomation was a viper of the *V. berus* subspecies. Moreover, other systemic manifestations rapidly resolved, and local symptoms progressively improved, until complete remission was achieved on the fifth day. No adverse reactions were observed after antivenom serum administration. This is consistent with the literature [2,4,6]. In our case, however, a slight increase in the IgE level, without clinical symptoms, was noted. The limitation of our study consisted of the identification modality of the snake, which was conducted by the herpetologist based on the following arguments: the viper is the only venomous snake in our country, the geographical mapping, and the morphological description provided by the adult accompanying the child at the time of the incident.

## 4. Conclusions

We can conclude that the venom of *V. berus* subspecies, found in the Carpathian Basin region, appears to have neurotoxic effects. This conclusion was reached from specialist consultation in *V. berus* identification, clinical observation, and possibly because of the complete and rapid remission of the symptoms under monospecific antivenom treatment. Consequently, we must take into consideration *V. berus* when confronted with neurological symptoms in patients bitten by snakes in the Subcarpathian Basin from Eastern Europe. However, further studies are necessary to confirm it.

## Figures and Tables

**Figure 1 toxins-14-00500-f001:**
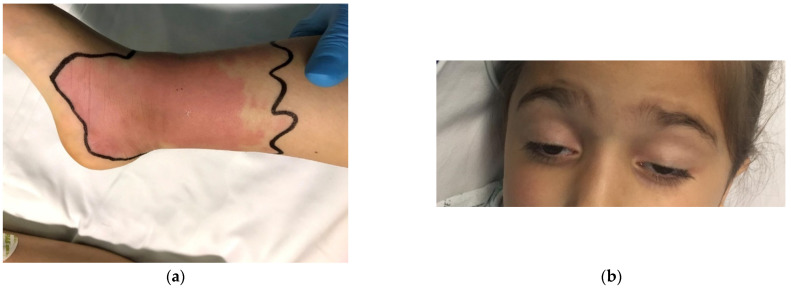
(**a**) Local signs of viper bite including swelling, erythema (lower calf and ankle); (**b**) image showing inferior displacement of the upper eyelid with associated narrowing of the vertical palpebral fissure (bilateral palpebral ptosis).

**Figure 2 toxins-14-00500-f002:**
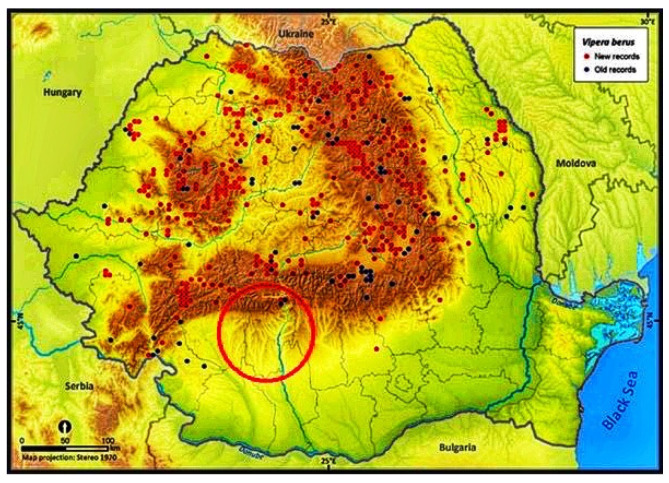
*V. berus* distribution in Romania [1]. Red dots: new records of *V. berus* after 1990; blue dots: records of *V. berus* before 1990.

**Table 1 toxins-14-00500-t001:** Laboratory results on days 1, 2, and 6.

Laboratory Test	Normal Values	Changes by Day:		
		Day 1 (admission)	Day 2	Day 6 (discharge)
Hemoglobin (g/dL)	12–16	11.8		13.2
Leucocytes (mmc)	4000–12,000	9100		5200
Thrombocytes (mmc)	150,000–400,000	321,000		293,000
ESR (mm/h)	7–12	7		
Fibrinogen (mg/dL)	150–400	243.9	253	
C reactive protein (mg/dL)	0–0.5	0.29		
ALT (IU/L)	0–35	24	21	27
AST (IU/L)	0–35	50	42	45
GGT (IU/L)	15–132		10	13
LDH (IU/L)	110–295	311	224	252
Amylase (IU/L)	22–80			39
Direct bilirubin (mg/dL)	0–0.2			0.06
Indirect bilirubin (mg/dL)	0–1			0.26
Total bilirubin (mg/dL)	0.3–1.2			0.32
Iron (μg/dL)	40–100			44
Chloride (mmol/L)	101–109	105	103	102
Sodium (mmol/L)	132–142	134	136	135
Potassium (mmol/L)	3.5–5.1	4.1	3.37	4.71
Urea (mg/dL)	10.8–38.4	28	28	23
Creatinine (mg/dL)	0.26–0.77	0.45	0.41	0.42
CK (IU/L)	0–145	120	106	135
CK-MB (ng/mL)	<5	<5	<1	<1
Myoglobin (ng/mL)	<50	<50	<50	<50
Troponin (ng/mL)	<1	<1	<0.05	<0.05
D-dimers (ng/mL)	<500		<100	<100
BNP (pg/mL)	<100		15.4	<5
IgA (g/L)	0.41–2.97			0.71
IgG (g/L)	5–13			10
IgM (g/L)	0.4–1.8			0.63
IgE (IU/mL)	<100	89.53		233
C3 fraction (g/L)	0.9–1.8			1.27
C4 fraction (g/L)	0.1–0.4			0.32
CIC (IU/mL)	<10			2
Quick Time (s)	11–14		14.1	13.1
INR	0.8–1.3		1.28	1.17
Prothrombin activity (%)	70–140		77.9	84.8
APTT (s)	24.4–36.4		25.1	21.6

ESR–erythrocyte sedimentation rate, ALT–alanine aminotransferase, AST–aspartate aminotransferase, GGT–gamma-glutamyl transferase, LDH–lactate dehydrogenase, CK–creatine kinase, CK-MB–creatine kinase–myocardial band, BNP–B-type natriuretic peptide, IgA–immunoglobulin A; IgG–immunoglobulin G, IgM–immunoglobulin M, IgE–immunoglobulin E, C3–complement C3 fraction, C4–complement C4 fraction, CIC–circulating immune complexes, INR–international normalized ratio, APTT–activated partial thromboplastin time.

**Table 2 toxins-14-00500-t002:** Audebert–Boels Classification modified by Marano et al. adapted from [6].

Grade	Description	Signs and Symptoms	Treatment
0	No envenoming (“dry bite”)	Fang marks No oedema No local reaction	- 6 h surveillance in the emergency room
1	Minimal envenoming	Local oedema around the bite area No systemic symptoms	- clinical observation up to evident reduction of edema - supportive care, including hydration and pain relief
2	Moderate envenoming	Grade 2a One or both of the following: - Regional edema with progression to most of the limb - Haematoma or adenopathy Grade 2b - Grade 2a + moderate general symptoms (mild hypotension, vomiting, diarrhea, neurotoxic signs), and/or biological criteria for severity: - Leukocytes > 11,000/L - Neutrophils > 65% - INR > 1.15	- Clinical observation up to the evident reduction of edema (evaluate district perfusion and saturation) - Supportive care, including hydration and pain relief - Doppler-ultrasound of affected limb’s blood vessels - Administration of antivenom - Evaluate antibiotic therapy * - Administer LMWH **
3	Severe envenoming	Other or both of the following: - Edema spreading to the trunk - Signs of hemodynamic instability (prolonged hypotension, shock, bleeding)	- Same intervention as in Grade 2 - Admission to PICU

* Only if clinical or laboratory signs of bacterial contamination are evident, ** Only if direct evidence of thrombophlebitis is available or in cases of extensive edema; dehydration; decreased mobility; prolonged decubitus; admission to PICU; anticipated hospitalization longer than 48 h. Do not administer in the case of overt hemorrhage or a bleeding disorder.

**Table 3 toxins-14-00500-t003:** Changes in the clinical profile of the patient over time.

Day	Clinical Features
Day 1: at the time of treatment initiation	Somnolence, palpebral ptosis, ophthalmoplegia, and bilateral diplopia. Mild gastrointestinal symptoms (nausea and diffuse abdominal pain). Local manifestations (swelling, erythema surrounded by cyanosis, local heat, induration, and pain in lower half of right calf). BP = 80/67 mmHg, HR = 116 beats/min
Day 1: 30 min after treatment initiation	Return of ocular movements. BP = 98/58 mmHg, HR = 118 beats/min
Day 2: 12 h after treatment initiation	No somnolence, no diplopia, and no gastrointestinal symptoms present. Persistence of palpebral ptosis. Local inflammation reduced. BP = 103/55 mmHg; HR = 80 beats/min
Day 3	Palpebral ptosis in remission, local signs improved (decreased swelling, no local heat, modest pain); BP = 90/66 mmHg; HR = 89 beats/min
Day 4	No palpebral ptosis noted. BP = 101/61 mmHg; HR = 105 beats/min
Day 5	No local signs or symptoms. BP = 107/67 mmHg; HR = 100 beats/min
Day 6	Complete remission was noted, and patient was discharged from our clinic.

BP = blood pressure; HR = heart rate.

## Data Availability

Not applicable.

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
