# Peer review of "Neurotoxicity and Other Clinical Manifestations of a Common European Adder (Vipera berus) Bite in Romania"

_toxins, 2022, doi:10.3390/toxins14070500_

Round 1

Reviewer 1 Report

Overall the information presented represents valuable information regarding the case study '' Neurotoxicity and other clinical manifestations of a common  European adder (Vipera berus) bite in Romania''. The paper is generally well written and structured. The authors should include a conclusion and recommendation for dealing with any future envenomation by this snake species. 

Author Response

Dear Reviewer,

Thank you for revising our manuscript entitled “  Neurotoxicity and other clinical manifestations of a common European adder (Vipera berus) bite in Romania” which we submitted to TOXINS Journal - Special Issue: Snakebite Clinics and Pathogenesis: From Preclinical to Resource Mapping Studies.

We  made changes to our manuscript taking into consideration your very useful  comments. Please see the attachment.

Sincerely yours, 

Reviewer 2 Report

All in all, an interesting read. A clear description of clinical symptoms of envenomation. Just a few things that I would like clarified or corrected:

Introduction

·         - Lines 29-31: A quick PubMed search revealed that clinical symptoms of neurotoxicity in Vipera berus bites have been observed as far back as 2008 (10.1093/qjmed/hcn079, https://doi.org/10.1016/j.neuro.2010.11.007, https://doi.org/10.1016/j.wem.2013.06.005).

Case report

·         - Lines 53-55: This is a pretty long sentence with a lot of “and”s. A nitpicky point, but it would benefit a reader to include commas here and/or break up this sentence.

·        -  Lines 56-59: Some context (or perhaps a table) would be helpful; how severe is a grade 2b on the scale?

·       -  Lines 60-64: I’m not entirely convinced here. According to the paper you cite on Romanian reptile distributions (and observations from iNaturalist), Vipera ammodytes is also found in Vâlcea county and is also a very similar size and colour. Can you be certain this bite is from Vipera berus?

Discussion

·         - Lines 127-132: Again, as someone unfamiliar with clinical snakebite grading systems, I’d like to know what these mean in wider context. Where do these grades place on the scale?

Author Response

Dear Reviewer,

Thank you for revising our manuscript entitled “Neurotoxicity and other clinical manifestations of a common European adder (Vipera berus) bite in Romania” which we submitted to TOXINS Journal - Special Issue: Snakebite Clinics and Pathogenesis: From Preclinical to Resource Mapping Studies.

We  made changes to our manuscript taking into consideration your very useful  comments. Please see the attachment.

Sincerely yours, 

Reviewer 3 Report

This is an interesting case describing neurotoxic symptoms after Vipera berus bites.
There are 3 vipers found in Romania, Vipera berus, V. ammodytes, V. ursinii. One publication about Vipera in Romania (Oleksandr Zinenko et al. 2010, Distribution and morphological variation of Vipera berus nikolskii ... Amphibia-Reptilia 31: 51-67) mentioned that a subspecies of Vipera berus, Vipera berus nikolskii. is also found in Romania and particularly it has been described in Vilcea, exactly the region where the girl had been bitten. Maybe the authors can contact the herpetologiste again, whether the responsible species could belong to this subspecies of V. berus. Maybe this particular subspecies causes neurotoxic symptoms. At least that could be discussed, as differentiation between the two species may be difficult. Furthermore the authors should change the legend of figure 2. According to the publication (reference 1) the red dots are new records after 1990 and the blue dots records before 1990.

Author Response

(The authors gave the same response as above.)

Reviewer 4 Report

A case report based on a potential envenoming from an unknown species of snake. The species was not identified but the suspected biting species is based on geographical location and a description from an unidentified source (potentially the 5 year old victim). The '.........complete resolution of symptoms of neurotoxicity after treatment with the monospecific antivenom..' (page 6) is not '.......evidence that the snake responsible for envenomation was a viper of the V. berus subspecies' (page 6) as suggested by the authors. Cross reactivity of antivenoms is well established. I do not believe that there is sufficient novel content to warrant publication.

Author Response

Dear Reviewer,

Thank you for revising our manuscript entitled “Neurotoxicity and other clinical of a common european adder (Vipera berus) bite in Romania” which we submitted to TOXINS Journal .

We  made changes to our manuscript taking into consideration your very useful  comments .Please see the attachment .

Sincerely yours, 

Round 2

Reviewer 4 Report

No further comments